# Deletion of Exon 1 in *AMER1* in Osteopathia Striata with Cranial Sclerosis

**DOI:** 10.3390/genes11121439

**Published:** 2020-11-30

**Authors:** Jingyi Mi, Padmini Parthasarathy, Benjamin J. Halliday, Tim Morgan, John Dean, Malgorzata J. M. Nowaczyk, David Markie, Stephen P. Robertson, Emma M. Wade

**Affiliations:** 1Department of Women’s and Children’s Health, Dunedin School of Medicine, University of Otago, Dunedin 9016, New Zealand; mije6460@student.otago.ac.nz (J.M.); mini.parthasarathy@otago.ac.nz (P.P.); halbe104@student.otago.ac.nz (B.J.H.); tim.morgan@otago.ac.nz (T.M.); emma.wade@otago.ac.nz (E.M.W.); 2North of Scotland Regional Genetics Service, NHS Grampian, Ashgrove House, Foresterhill, Aberdeen AB25 2ZA, UK; j.dean@abdn.ac.uk; 3Department of Pathology and Molecular Medicine, Health Sciences Centre, McMaster University, Hamilton, ON L8S 4K1, Canada; nowaczyk@hhsc.ca; 4Department of Pathology, Dunedin School of Medicine, University of Otago, Dunedin 9016, New Zealand; david.markie@otago.ac.nz

**Keywords:** *AMER1*, *WTX*, osteopathia striata with cranial sclerosis

## Abstract

Osteopathia striata with cranial sclerosis (OSCS) is an X-linked dominant condition characterised by metaphyseal striations, macrocephaly, cleft palate, and developmental delay in affected females. Males have a more severe phenotype with multi-organ malformations, and rarely survive. To date, only frameshift and nonsense variants in exon 2, the single coding exon of *AMER1*, or whole gene deletions have been reported to cause OSCS. In this study, we describe two families with phenotypic features typical of OSCS. Exome sequencing and multiplex ligation-dependent probe amplification (MLPA) did not identify pathogenic variants in *AMER1*. Therefore, genome sequencing was employed which identified two deletions containing the non-coding exon 1 of *AMER1* in the families. These families highlight the importance of considering variants or deletions of upstream non-coding exons in conditions such as OSCS, noting that often such exons are not captured on probe or enrichment-based platforms because of their high G/C content.

## 1. Introduction

Osteopathia striata with cranial sclerosis (OSCS: MIM# 300373) is a rare skeletal dysplasia caused by loss-of-function or truncating variants in, or deletion of, *AMER1* (MIM# 300647; also known as *WTX/FAM123B*) [1,2,3]. OCSC has an X-linked, dominant mode of inheritance and, in females, is primarily characterised by metaphyseal striations, osteosclerosis, macrocephaly, hypertelorism and occasionally, developmental delay. Affected males rarely survive the perinatal period; the occasional survivor exhibits osteosclerosis without metaphyseal striations, as well as cardiovascular, gastrointestinal, and genitourinary malformations, alongside features present in typically affected females. Males with mosaicism for loss of function *AMER1* variants, or those with Klinefelter syndrome (XXY), manifest a phenotype similar to female heterozygotes [4].

*AMER1* encodes adenomatous polyposis coli (APC) membrane recruitment protein 1, which regulates the WNT1 signalling pathway by stimulating *β*-catenin degradation. Loss-of-function variants resulting in reduced expression of *AMER1* cause enhanced WNT signalling through increased nuclear accumulation of *β*-catenin [5]. WNT signalling is involved in cellular differentiation and proliferation during embryogenesis and tumorigenesis, hence its loss results in a pleiotropic clinical phenotype. Increased WNT-signalling causes enhancement of osteoblast maturation and function [6,7], leading to increased bone density [8].

In this report, we describe two families with OSCS phenotypes who were found to have novel pathogenic variants in *AMER1*, (NM_152424.4:c.-417_-99+1606del) and (NM_152424.4:c.-1290_-99+380del), resulting in isolated deletion of the 5’ non-coding exon 1. Previous attempts at a diagnosis using exome sequencing and multiplex ligation-dependent probe amplification (MLPA) did not detect the deletions because these platforms did not target this non-coding exon for sequencing and dosage analysis respectively. The deletions were subsequently discovered using genome sequencing (GS). These cases highlight the importance of screening 5′ untranslated exons in genes that produce phenotypes due to a loss-of-function mechanism.

## 2. Materials and Methods

### 2.1. Subjects and Ethical Consent

Two unrelated individuals with OSCS were ascertained by physician-initiated referral. The study was conducted in accordance with the Declaration of Helsinki and families were consented to participate in a research study under approved protocols MEC/08/08/094 and 13/STH/56 (Health and Disability Ethics Committee, Dunedin, New Zealand). Clinical images are reproduced with consent from the families.

### 2.2. Sequence Analysis

Genomic DNA was extracted from blood from the affected individuals and their family members. X-inactivation analysis, and multiplex ligation-dependent probe (MLPA) analysis and targeted Sanger sequencing of the *AMER1* coding exon were performed as described previously [9].

Exome sequencing was performed by Otogenetics (Atlanta, GA, USA), and genome sequencing was performed by the Kinghorn Centre for Clinical Genomics (Sydney, Australia) using a HiSeq2500 (Illumina, San Diego, CA, USA). Exome enrichment was performed with the Agilent SureSelect Human All Exon V4+UTRs (untranslated regions) or V5+UTRs capture kits (Agilent Technologies, Santa Clara, CA, USA). Preparation for genome sequencing was performed with the Illumina 30x GS (TruSeq Nano (Illumina)) v2.5 kit. Reads were paired, with a read length of 150 base pairs. Alignment of reads and variant calling was done according to the Genome Analysis Tool Kit (GATK) best practice guidelines [10,11]. Alignment to the reference sequence (GRCh37 assembly) was performed with the Burrows-Wheeler Aligner (v0.7.17) with the MEM algorithm [12]. Picard MarkDuplicates (v2.18.11) was used to identify duplicate reads [13] and aligned files in bam format were produced using GATK (v3.8) BaseRecalibrator. GATK HaplotypeCaller was used to generate GVCFs, containing single nucleotide variants (SNVs) and short insertions/deletions (indels). Multi-sample VCFs were produced using GATK GenotypeGVCFs, which were annotated with gene context information using SnpEff (v4.3S) [14]. Population allele frequencies from the gnomAD project (v2.1.1) [15] were added using BCFtools annotate (v1.9).

Manta (v1.6.0) [16] was used to call large deletions and duplications using the genome sequences, as well as inversion and insertion events and SnpEff was used to annotate the structural events for genic impact, in VCF format. Data from the Database of Genomic Variants (DGV, release date 15 May 2016) was used to identify novel structural events.

Allele specific PCR using genomic DNA extracted from dermal fibroblasts confirmed segregation of the variant within the families. PCR was performed with Q5 HotStart High Fidelity DNA polymerase (NEB, Ipswich, MA, USA). Individual PCR primers were designed for the two families, but the experimental design was the same. Primer set 1 was designed to flank the deletions, producing products too large for amplification under the chosen conditions from the WT allele, therefore amplifying only the deletion-containing allele, producing products of 1221 bp (Family A) or 546 bp (Family B). Primer set 2 consists of one primer embedded within the deletion, amplifying only the wild-type allele, producing products of 662 bp (Family A) or 669 bp (Family B). PCR conditions were appropriate for amplification of 100–1200 bp products. Primer sequences are available upon request.

## 3. Results

### 3.1. Clinical Descriptions

#### 3.1.1. Proband A

Proband A was diagnosed with bilateral cleft lip and palate, and bilateral absence of fibulae on prenatal ultrasound at 26th week gestation. She was born at 30 weeks and 4 days. She had an abnormal head shape, hypertelorism, bilateral cleft lip and palate, and bilateral choanal atresia. She also manifested severe features more commonly seen in the male OCSC phenotype including ventricular septal defect, a secundum atrial septal defect, patent ductus arteriosus, hydronephrosis, intestinal malrotation, and anal stenosis. She required a tracheostomy for laryngotracheomalacia. Magnetic Resonance Imaging (MRI) of the head showed hypoplasia of the splenium of the corpus callosum. Radiographs showed generalized increased bone density, absent fibulae bilaterally, and shortening and undermodelling of the proximal phalanges (Figure 1A–C). Chromosomal microarray demonstrated a normal female chromosomal complement. A clinical diagnosis of OSCS was made in the proband. Her father had a phenotype that included metaphyseal striations, generalised skull osteosclerosis, macrocephaly, coarse facial features, and thickening of the nasal bridge.

#### 3.1.2. Proband B

Proband B was born at 35 weeks gestation by caesarean section because of polyhydramnios. Bilateral cleft lip and palate, macrocephaly, hypertelorism, and lymphoedema were noted at birth. She had a nasopharyngeal airway inserted because of upper airway obstruction. She had a bilateral cleft lip repair at 1 year 11 months, and cleft palate repair at 2 years 6 months. A skeletal survey demonstrated metaphyseal striations (Figure 1D) with calvarial thickening. There was no family history. A clinical diagnosis of OSCS was made. Although she had delayed speech initially, her neurodevelopment was subsequently within normal limits, and at age 7 years, she is making normal progress in mainstream education, but continues to need nocturnal continuous positive airway pressure (CPAP).

### 3.2. Non-Coding Deletions in AMER1 Cause OSCS

*AMER1* (Xq11.2) contains 2 exons. Exon 1 is non-coding and exon 2 contains the translation initiation site for the coding region. In proband A, Sanger sequencing and MLPA analysis of the single coding exon of *AMER1* was performed, and no pathogenic variants were found. There was no skewing of the ratio of X-chromosome inactivation (54:46). The same sequence and MLPA analysis of *AMER1* was performed in her father and demonstrated no anomalies. Similarly, in proband B, no pathogenic variants were found by Sanger sequencing or MLPA analysis. The absence of discoverable variants in *AMER1* led us to seek pathogenic variants at another locus and whole exome sequencing was carried out. This approach failed to identify plausible candidate variants. Since the exon capture platforms used here did not include the first non-coding exon in *AMER1*, and due to the potential for locus heterogeneity underlying OSCS, whole genome sequencing was utilised. In both probands, a deletion of the 5’ non-coding exon which encodes most of the 5′ untranslated region of *AMER1* was identified. Proband A was heterozygous for a 1924 bp deletion, (NM_152424.4:c.-417_-99+1606del, NC_000023.10:g.63423844_63425768del), encompassing the entirety of exon 1 (Figure 1B). PCR analysis (Figure 1C,D) showed that the paternal grandmother, paternal aunt and mother amplified the reference allele. The proband and her father produced both a product amplified from the reference allele and a product amplified by the internal primers (deletion allele). This indicates that the proband has one deletion-containing and one reference allele present. Given that both PCR reactions produced products in the father (A-II-2), it is concluded that the deletion arose early in his development rendering him mosaic for the variant which he then transmitted to the proband.

Proband B was shown to be heterozygous for a de novo 1571 bp deletion in *AMER1*, (NM_152424.4:c.-1290_-99+380del, NC_000023.10:g.63425070_63426641del), encompassing the entirety of exon 1 that encodes most of the 5′ untranslated region of *AMER1* (Figure 1F). PCR results (Figure 1G,H) confirmed that both parents only produced the reference product, indicating that they are hemizygous and homozygous, respectively, for the reference allele. The proband (B-II-1) amplified products for both pair of primers, indicating that she carries both a reference and the deletion-containing allele. The pathogenic variant therefore arose de novo in the proband. These variants were not detected using MLPA because a probe pair was not designed for this region of the gene.

Both deletions encompass the entirety of exon 1 (chrX:63,425,624_63,425,450, GRCh37.p13), delete the sole transcription start site for the gene, and predict a loss of transcript produced from this allele. Both have been assigned ‘Pathogenic’ as per American College of Medical Genetics guidelines (PVS1, PM2, PM4, PP1, PP3 and PP4) [17] and have been submitted to ClinVar (accessions SCV001426685 and SCV001426686, respectively).

## 4. Discussion

Germline truncating variants in *AMER1*, as well as whole gene deletions, cause OSCS [3,9]. In this study, two different deletions of the 5′ non-coding exon were found in two families. In family A, the deletion segregated with the phenotype; in family B, the deletion arose de novo in the proband. Possible mechanisms of pathogenicity could include: deletion of the transcription start site such that RNA polymerase cannot bind and initiate transcription, or deletion of the splice sites flanking exon 1 resulting in abnormal transcript maturation. In both instances, functional protein will not be produced and both alleles can be considered to be null.

Despite being non-coding, loss of *AMER1* exon 1 results in the same phenotype as loss-of-function variants in the coding region of *AMER1*, or whole-gene deletions of the locus [9]. The pathogenic deletions were not discovered by conventional diagnostic methods. Initial Sanger sequencing of *AMER1* did not target exon 1 as variants in this region had not been previously implicated in OSCS. MLPA did not detect the deletions, because of an inability to locate a probe pair in such as small sized exon due to its sequence composition. In whole exome sequencing, the 5′ *AMER1* non-coding exon was not represented on the capture platforms used in this study and therefore not sequenced. However, it should be noted that more recent capture kits that capture UTRs do target *AMER1* exon 1, for example Agilent SureSelect Human All Exon V6/V7+UTRs, however the high G/C content may still prevent satisfactory sequencing of the region.

These two cases demonstrate the need to conduct similar, focused searches in other genes associated with loss-of-function phenotypes that contain upstream non-coding exons. Deletions that encompass 5′ non-coding exons may be frequently overlooked in conventional diagnostic methods for a number of reasons. Upstream, non-coding exons are often small, therefore deletions that remove them can also be small and likely to be missed by large-scale genomic methods to detect copy number variants. They are also commonly separated from the first coding exon by large introns, therefore targeted sequencing of the whole gene may not extend to these regions, or they will not be captured in some exome sequencing protocols. Furthermore, the frequently high G/C content of these regions complicates the design of efficient capture probes for exome or MLPA type-analyses. Consequently, although focused sequencing may detect single nucleotide variants in 5′ UTRs, a targeted approach to identify pathogenic deletions outside of the coding region should also be considered.

## 5. Conclusions

In conclusion, we identified two novel variants in *AMER1*, leading to deletion of the 5′ non-coding exon, in two families with OSCS. This demonstrates that the whole gene should be considered when screening for OSCS, not merely the coding region of exon 2. Inclusion of methodologies such as whole genome sequencing and PCR can increase diagnostic sensitivity in conditions with similar circumstances to what we describe here for OSCS.

## Figures and Tables

**Figure 1 genes-11-01439-f001:**
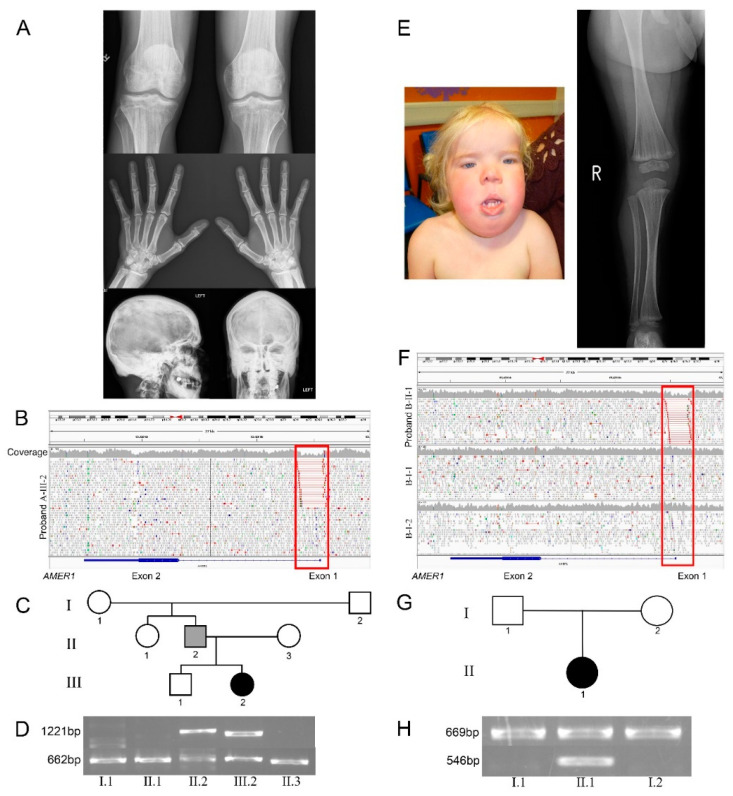
Clinical and sequence data from families A and B. (**A**) Radiographs from the proband of family A showing metaphyseal striations, cranial sclerosis, and osteosclerosis in the proband; (**B**) Integrated genome viewer (IGV) screen-shot illustrating the deletion NC_000023.10:63423844_63425768, encompassing exon 1, in the proband (red box). (**C**) Pedigree and (**D**) PCR results from family A showing individuals I-1, II-2 and II-3 have a band of 662 bp corresponding to DNA amplified by primer pair 2, the proband III-2 and her father II.2 have bands of both 662 bp and 1221bp corresponding to DNA amplified by both sets of primers. (**E**) Clinical image and radiographs from the proband of family B showing macrocephaly, hypertelorism and metaphyseal striations. (**F**) An IGV screen-shot showing a de novo deletion NC_000023.10:63425070_63426641 in the proband (red box). (**G**) Pedigree and (**H**) PCR results are consistent with both parents being unaffected, each carrying only the wild type allele and producing only the primer set 2 product (669 bp), II-1 is heterozygous for a de novo deletion with 2 bands indicating amplification by both primer sets (primer set 1: 546 bp, primer set 2: 669 bp).

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
