# Peer review of "Deletion of Exon 1 in AMER1 in Osteopathia Striata with Cranial Sclerosis"

_genes, 2020, doi:10.3390/genes11121439_

Round 1
Reviewer 1 Report
This short report describes the detection of two novel deletions of the non-coding exon 1 of AMER1, which were only detected when WGS was undertaken. This case clearly demonstrates the utility of WGS v WES and MLPA.
Comments
- There appears to be an error in the description of the PCR in the text compared to what is seen in the gel in Fig 1D and 1H and the corresponding legend. In the gel Family A has two bands of 1221bp (del) and 662bp (WT) but it is stated in the text as 546bp and 662pb respectively. Family B has two bands of 669pb (WT) and 546bp (del) but in the text it says 1221pb and 669bp respectively.
Text: “Primer set 1 was designed to flank the deletions, producing products too large for amplification under the chosen conditions from the WT allele, therefore amplifying only the deletion-containing allele, producing products of 546 bp (Family A) or 1221 bp (Family B). Primer set 2 consists of one primer embedded within the deletion, amplifying only the wild-type allele, producing products of 662 bp (Family A) or 669 bp (Family B).”
Fig 1D legend: “PCR results from family A showing individuals I-1, II-2 and II-3 have a band of 662 bp corresponding to DNA amplified by primer pair 2, the proband III-2 has bands of both 662 bp and 1221bp corresponding to DNA amplified by both sets of primers, individual II-2 shows a band of 1221 bp corresponding to DNA amplified by primer set 1 plus a faint band corresponding to DNA amplified by primer set 2 (662 bp), consistent with mosaicism.”
Fig 1H legend: “PCR results are consistent with both parents being unaffected, each carrying only the wild type allele and producing only the primer set 2 product (669 bp), II-1 is heterozygous for a de novo deletion with 2 bands indicating amplification by both primer sets (primer set 1: 546 bp, primer set 2: 669 bp).”
- It would also be better to redraw family A pedigree so that II.2 and III.1 are not directly above and below each other. Also it would help if they are drawn above the corresponding gel lanes. Also please indicate the pedigree numbers above the lanes in Fig 1D and 1H so as to avoid having to read the figure legends.
- Results – molecular details. This section could be reduced, there is too much detail on describing the PCRs of the WT and deletion in the results. Its described in the methods and in the figure 1 legend.
- From the gel in Fig 1D, the mosaicism present in II.2 does not look that low. It’s a shame that WGS was not performed so that an estimate could have been made.
- Results line 155: The authors describe that the mosaicism as a somatic event but this can lead to confusion as the variant was also in his germinal cells as his daughter inherited the deletion. This could be made a bit clearer.
- Results line 157 – Family B - they describe the PCRs in detail showing that the deletion is de novo but don´t mention that the WGS data shows this too.
Author Response
Thank you for your thorough and thoughtful review, we have addressed your concerns as laid out below.
- There appears to be an error in the description of the PCR in the text compared to what is seen in the gel in Fig 1D and 1H and the corresponding legend. In the gel Family A has two bands of 1221bp (del) and 662bp (WT) but it is stated in the text as 546bp and 662pb respectively. Family B has two bands of 669pb (WT) and 546bp (del) but in the text it says 1221pb and 669bp respectively.
Text: “Primer set 1 was designed to flank the deletions, producing products too large for amplification under the chosen conditions from the WT allele, therefore amplifying only the deletion-containing allele, producing products of 546 bp (Family A) or 1221 bp (Family B). Primer set 2 consists of one primer embedded within the deletion, amplifying only the wild-type allele, producing products of 662 bp (Family A) or 669 bp (Family B).”
Fig 1D legend: “PCR results from family A showing individuals I-1, II-2 and II-3 have a band of 662 bp corresponding to DNA amplified by primer pair 2, the proband III-2 has bands of both 662 bp and 1221bp corresponding to DNA amplified by both sets of primers, individual II-2 shows a band of 1221 bp corresponding to DNA amplified by primer set 1 plus a faint band corresponding to DNA amplified by primer set 2 (662 bp), consistent with mosaicism.”
Fig 1H legend: “PCR results are consistent with both parents being unaffected, each carrying only the wild type allele and producing only the primer set 2 product (669 bp), II-1 is heterozygous for a de novo deletion with 2 bands indicating amplification by both primer sets (primer set 1: 546 bp, primer set 2: 669 bp).”
-
- Apologies for this error, on review it seems the mistake is in the methods section and has been corrected (lines 88 and 89)
- It would also be better to redraw family A pedigree so that II.2 and III.1 are not directly above and below each other. Also it would help if they are drawn above the corresponding gel lanes. Also please indicate the pedigree numbers above the lanes in Fig 1D and 1H so as to avoid having to read the figure legends.
- We agree that the figure could be clearer, and the pedigree has been redrawn. Pedigree numbers have also been added to the figure.
- Results – molecular details. This section could be reduced, there is too much detail on describing the PCRs of the WT and deletion in the results. Its described in the methods and in the figure 1 legend.
- We agree the PCR description is repetitive and this section has been reduced for clarity.
- From the gel in Fig 1D, the mosaicism present in II.2 does not look that low. It’s a shame that WGS was not performed so that an estimate could have been made.
- You are right, on another close look, the band is just as bright as his daughters. We have removed description of the mosaicism as "low" and not made comment on how many cells may carry this deletion in the father.
- Results line 155: The authors describe that the mosaicism as a somatic event but this can lead to confusion as the variant was also in his germinal cells as his daughter inherited the deletion. This could be made a bit clearer.
- Thank you for your suggestion to clarify, we have edited the sentence to read "it is concluded that the deletion arose early in his development rendering him mosaic for the variant which he then transmitted to the proband."
- Results line 157 – Family B - they describe the PCRs in detail showing that the deletion is de novo but don´t mention that the WGS data shows this too.
- The results section describing the findings in family B (lines 161 to 169) has been updated to include conclusions from both the WGS and the PCR data
Reviewer 2 Report
The manuscript “Deletion of Exon 1 in AMER1 in Osteopathia Striata 2 with Cranial Sclerosis” describes new deletions in AMRE1 gene leading to loss-of-function in two families with phenotypic features typical of Osteopathia striata with cranial sclerosis (OSCS).
This manuscript is clearly presented with clinical and pathophysiological data, a brief introduction of the biological function of AMER1 in the WNT pathway and sequencing results.
I suggest only few improvements for readers.
- In the abstract, authors should introduce that AMER1 gene is made of 2 exons. Please consider to replace “To date, only frameshift and nonsense variants in the single coding exon of AMER1,” by “To date, only frameshift and nonsense variants in the exon 2, the single coding exon of AMER1,”. (line 19)
- In the introduction, authors should precise that the 5’ untranslated regions that are deleted in the two presented families, are located within exon 1 of AMER1. Indeed a short 5’ untranslated region is contained within exon 2 of AMER1, as well specified by authors between lines 134 to 147, but readers should understand this earlier. Please consider to replace “resulting in isolated deletion of the 5’ non-coding exon” by “resulting in isolated deletion of the 5’ non-coding exon 1”. (line 48)
- Latin words should appear in italics, such as corpus callosum…
- Full name is needed for “CPAP” (continuous positive airway pressure) (Line 115)
Author Response
Thank you for your thorough and thoughtful review, our corrections are laid out below:
- In the abstract, authors should introduce that AMER1 gene is made of 2 exons. Please consider to replace “To date, only frameshift and nonsense variants in the single coding exon of AMER1,” by “To date, only frameshift and nonsense variants in the exon 2, the single coding exon of AMER1,”. (line 19)
- Thank you for suggesting this clarification, line 19 now reads: "To date, only frameshift and nonsense variants in exon 2, the single coding exon of AMER1, or whole gene deletions have been reported to cause OSCS"
- In the introduction, authors should precise that the 5’ untranslated regions that are deleted in the two presented families, are located within exon 1 of AMER1. Indeed a short 5’ untranslated region is contained within exon 2 of AMER1, as well specified by authors between lines 134 to 147, but readers should understand this earlier. Please consider to replace “resulting in isolated deletion of the 5’ non-coding exon” by “resulting in isolated deletion of the 5’ non-coding exon 1”. (line 48)
- Again, we agree this should be clarified and line 48 now reads "), resulting in isolated deletion of the 5’ non-coding exon 1"
- Latin words should appear in italics, such as corpus callosum…
- Italicizing latin terms is not the practice of this journal, so we have left these terms in standard font.
- Full name is needed for “CPAP” (continuous positive airway pressure) (Line 115)
- Thank you for pointing this out, corrected